# Pharmacokinetics of Toxin-Derived Peptide Drugs

**DOI:** 10.3390/toxins10110483

**Published:** 2018-11-20

**Authors:** David Stepensky

**Affiliations:** Department of Clinical Biochemistry and Pharmacology, Faculty of Health Sciences, Ben-Gurion University of the Negev, P.O. Box 653, Beer-Sheva 84105, Israel; davidst@bgu.ac.il; Tel.: +972-8647-7381

**Keywords:** toxin-derived peptide drugs, physicochemical properties, pharmacokinetics, pharmacokinetic parameters, pharmacokinetic variability factors, dose adjustment

## Abstract

Toxins and venoms produced by different organisms contain peptides that have evolved to have highly selective and potent pharmacological effects on specific targets for protection and predation. Several toxin-derived peptides have become drugs and are used for the management of diabetes, hypertension, chronic pain, and other medical conditions. Despite the similarity in their composition (amino acids as the building blocks), toxin-derived peptide drugs have very profound differences in their structure and conformation, in their physicochemical properties (that affect solubility, stability, etc.), and subsequently in their pharmacokinetics (the processes of absorption, distribution, metabolism, and elimination following their administration to patients). This review summarizes and critically analyzes the pharmacokinetic properties of toxin-derived peptide drugs: (1) the relationship between the chemical structure, physicochemical properties, and the pharmacokinetics of the specific drugs, (2) the major pharmacokinetic properties and parameters of these drugs, and (3) the major pharmacokinetic variability factors of the individual drugs. The structural properties of toxin-derived peptides affect their pharmacokinetics and pose some limitations on their clinical use. These properties should be taken into account during the development of new toxin-derived peptide drugs, and for the efficient and safe use of the clinically approved drugs from this group in the individual patients.

## 1. Introduction

Peptides are an important component of toxins and venoms produced by different organisms. They evolved to have highly selective and potent pharmacological effects on specific targets for protection and predation. These properties, and advances in the detection, analytics, and synthesis of peptides and their derivatives, make them promising leads for the development of new drugs [1]. Indeed, several toxin-derived peptides have become drugs for the management of diabetes, hypertension, chronic pain, and other medical conditions [2]. Dozens of toxin-derived peptides and their derivatives are undergoing clinical trials or are in the pre-clinical development stages.

Despite the similarity in their composition (amino acids as the building blocks), toxin-derived peptides have very profound differences in their structure and conformation. Only part of this “chemical space” can apparently lead to successful drug leads and new drugs. This is because the unfavorable physicochemical and pharmacokinetic (PK) properties of a drug candidate, even if it possesses a highly potent and selective desired pharmacological activity, can lead to failure in the pre-clinical and clinical development, or/and to substantial limitations in its clinical use.

The objective of this review is to critically summarize and analyze the pharmacokinetic properties of toxin-derived peptide drugs. Specifically, (1) to describe the relationship between the chemical structure, physicochemical properties, and the pharmacokinetics of the specific drugs, (2) summarize the major pharmacokinetic properties and parameters of these drugs, (3) to describe the major variability factors and extent of their effect on the pharmacokinetics of the individual drugs.

The focus of this review is on the drugs that were approved for clinical use (see Table 1 for the details of the analyzed drugs, including their origin, indication, and innovative drug products), and not on the agents that are evaluated currently in pre-clinical or clinical trials, or have been discontinued.

## 2. Structure and Physicochemical Properties of Clinically-Approved Toxin-Derived Peptide Drugs

The chemical structures of the clinically approved toxin-derived peptide drugs are shown in Figure 1. It can be seen that there are profound differences in the structures of the individual agents: composition, size, the presence of intramolecular bonds/rings, conformation, etc. As a result, these agents possess very different physicochemical properties, some of which are summarized in Table 2. Specifically, the molecular weight of the drugs varies profoundly between 217–4910 g/mol (please compare to the commonly-approved cutoff of 1000 g/mol for the small molecular weight (MW)-drugs). These drugs exist in numerous conformations, and for some of the molecules, the conformational space is limited by the cyclic structure or the presence of disulfide intramolecular bridges (see Figure 1).

These analyzed compounds contain in their sequence hydrophobic and hydrophilic amino acids, and the later ones (such as carboxylic acid and amine groups) can be charged at the physiological pH (see Figure 1 and Table 2). Overall, there are profound differences in the chemical structure of the analyzed compounds that affect the conformation, solubility, and stability of the specific compound, and its ability to interact with endogenous compounds (e.g., albumin and metabolic enzymes) and permeate biological membranes (e.g., the gastrointestinal wall or other biological barriers). In the next section, we will analyze the effect of the structural and physicochemical properties (see Figure 1 and Table 2) on the pharmacokinetics of the individual agents (see Table 1), and the sources of pharmacokinetic variability in the individual patients that are treated with these drugs.

## 3. Pharmacokinetic Properties of the Clinically Approved Toxin-Derived Peptide Drugs

### 3.1. Captopril

Captopril is a small molecular weight drug that has been used clinically from the 1980s. It was the first orally active inhibitor of the angiotensin-converting enzyme, and it is still used widely in the treatment of hypertension and congestive heart failure [2].

Due to its chemical structure and physicochemical properties, captopril has good solubility in water and gastrointestinal (GI) fluids (see Table 2). However, its absorption is limited by its low permeability through the biological membranes (including the GI wall), and captopril is classified into Class III (high solubility, low permeability) [3] of the biopharmaceutics classification system (BCS) [4]. Overall, about 60% to 75% of an oral dose of captopril is absorbed (see Table 3), and its peak blood concentrations are reached about 45 min to 60 min after oral administration [5,6]. Co-administration with food decreases the bioavailability of captopril by 30–40%, and it is recommended to be given one hour before meals [5]. However, the decreased bioavailability of captopril when taken with meals does not significantly alter clinical responses to the drug [6].

Captopril contains a sulfhydryl group and binds to albumin and endogenous thiol-containing compounds (such as cysteine and glutathione) in the plasma and other body fluids. It also exists in a form of a disulfide dimer of the parent drug. These compounds (that are described collectively as “total captopril”) serve as a reservoir of the pharmacologically-active drug, and prolong its pharmacological effect. Overall, approximately 30% of the captopril in the central circulation is bound to the plasma proteins, and its volume of distribution is 0.8 L/kg [6] (that corresponds to 56 L in a 70-kg patient).

The clearance of captopril in healthy subjects is approximately 49 L/h, resulting in an elimination half-life of unchanged captopril of approximately 2 h. Captopril is eliminated from the body primarily by the kidneys, and undergoes active secretion in the tubuli of the nephron [5,6]. Decreased renal function leads to accumulation of the drug in the body and necessitates dose reduction [5,7] (see Table 4). Similar phenomenon and the need for dose reduction are observed when captopril is administered with probenecid, which is apparently due to competition between these drugs for tubular secretion in the kidneys [6]. Overall, the pharmacokinetics of captopril appear to be preserved in patients with hypertension and congestive heart failure, and are not affected by many concomitant medications [7,8].

### 3.2. Cyclosporine

Cyclosporine is a cyclic polypeptide immunosuppressant agent consisting of 11 amino acids that is produced by the fungus species *Beauveria nivea* (see Table 1 and Figure 1). It is indicated for the prophylaxis of organ rejection in kidney, liver, and heart allogeneic transplants, and in some patients with rheumatoid arthritis and psoriasis [9]. Cyclosporine was introduced into the clinical practice in the early 1980s, and it has been gradually replaced in the last few years by the newer agents [10] (such as tacrolimus, sirolimus, therapeutic antibodies, and other drugs).

Cyclosporine is classified as a BCS class II (low solubility, high permeability) drug [3]. Following oral administration, the absorption of cyclosporine is incomplete, and the absolute bioavailability generally varies between 10–89%, and is dependent on the individual patient, the patient population, the formulation, concomitant treatments, diet, etc. Cyclosporine undergoes substantial first-pass metabolism in the GI wall and in the liver by the CYP3A4 enzyme. It is also a substrate of the P-glycoprotein (Pgp) and other efflux pumps (that are expressed in the GI wall, liver, kidneys, and other locations in the body) [10].

Cyclosporine distributes extensively in the body, and the steady-state volume of distribution is 3–5 L/kg. The drug distribution in the blood is approximately 41% to 58% in erythrocytes, 33% to 47% in plasma, 5% to 12% in granulocytes, and 4% to 9% in lymphocytes [11]. In plasma, cyclosporine binds primarily to lipoproteins and secondarily to albumin. Its unbound fraction in the plasma equals approximately 10%. The disposition of cyclosporine from blood is generally biphasic, with a terminal half-life of approximately 8.4 h (range 5–27 h) [9,11].

The elimination of cyclosporine is primarily biliary with only 6% of the dose (parent drug and metabolites) excreted in urine. The clearance of cyclosporine is approximately 5–7 mL/min/kg in adult recipients of renal or liver allografts, and slightly lower in cardiac transplant patients. Substrates, inhibitors, and inducers of the CYP3A enzyme affect the bioavailability and the clearance of cyclosporine [9]. Therefore, dozens of drugs and nutrients have pharmacokinetic interaction with cyclosporine, affect its concentrations, and can lead to inefficiency or the excessive toxicity of this drug.

The substantial intersubject and intrasubject pharmacokinetic variability of cyclosporine, in combination with its narrow therapeutic window, necessitate dosage adjustment based on the therapeutic monitoring of its trough blood concentrations [9]. Body weight, the co-administration of inhibitors of CYP3A, and hematocrit appear to be the most important factors associated with the clearance of cyclosporine in transplant patients, which necessitate dose adjustment [11,12] (see Table 4). The genetic effects (e.g., CYP3A4 variant alleles) appear to have a minor effect on the pharmacokinetics of cyclosporine, and the genotyping of the patients is rarely applied to individualize and optimize the dosing of this drug [13,14].

### 3.3. Eptifibatide

Eptifibatide is a cyclic heptapeptide that reversibly inhibits platelet aggregation by preventing the binding of fibrinogen, von Willebrand factor, and other adhesive ligands to glycoprotein IIb/IIIa. Eptifibatide is indicated in patients with acute coronary syndrome and percutaneous coronary intervention (PCI), and the recommended dosing regimen is IV bolus of the drug followed by an IV infusion, at the doses adjusted by the patients’ weight [15].

The volume of distribution of eptifibatide in patients with coronary artery disease is about 185–260 mL/kg, and is somewhat higher (220–270 mL/kg) in healthy individuals [16]. The drug in the plasma is present largely in an unbound form, and the plasma elimination half-life is approximately 2.5 h.

Clearance in patients with coronary artery disease is 55–58 mL/kg/h [15]. In healthy subjects, renal clearance accounts for approximately 50% of total body clearance, with the majority of the drug excreted in the urine as the original drug or its metabolites [17]. Moderate to severe renal impairment resulted in a ~50% reduction in total eptifibatide clearance and a corresponding doubling of plasma eptifibatide concentration [18]. Therefore, it was recommended to reduce twofold the IV infusion component of eptifibatide in these patients. The IV bolus component of the dose is used to establish the desired (steady-state) eptifibatide concentrations, is not based on the value of the drug clearance (but rather on the volume of drug distribution), and should not be altered [18].

The pharmacokinetics of eptifibatide apparently is not affected by the gender, but there are indications that the clearance declines with age (in the geriatric versus adult subjects) [15].

### 3.4. Lepirudin

Lepirudin is a recombinant 65-aa hirudin derived from yeast cells, which bears some differences in its chemical structure from the natural hirudin [19]. It binds to thrombin, blocks its thrombogenic activity, and is used for anticoagulation in patients with heparin-associated thrombocytopenia (HIT) and associated thromboembolic disease. The marketing of lepirudin was discontinued in 2012, apparently due to the commercial reasons.

Lepirudin is recommended to be administered by IV bolus followed by IV infusion [19], but subcutaneous administration of the drug and its subsequent pharmacokinetics were also characterized [20]. Following intravenous administration, the time course of lepirudin plasma concentrations follows a two-compartment model, with an initial half-life of approximately 10 min, and a terminal half-life of about 1.3 h in young healthy volunteers [19,20]. At the steady state, the drug distribution is confined to the extracellular fluids (V = 12.2 L in the healthy young subjects). The volume of distribution of lepirudin increases to ~18 L in the elderly patients and in patients with renal impairment (CLcr, creatinine clearance <80 mL/min), and increases substantially (32.1 L; highly variable parameter, %CV = 98.9%) in patients with heparin-induced thrombocytopenia [19].

Lepirudin is eliminated by the kidneys (48% of the IV dose, mostly in the form of unchanged drug) and apparently also undergoes catabolic hydrolysis (proteolysis in the plasma and extracellular fluids) [20]. Clearance of the drug is proportional to the kidney function, and proportional dosage reduction based on creatinine clearance is recommended in patients with renal impairment [19]. In patients with heparin-induced thrombocytopenia, the drug clearance is reduced by 20–30%, as compared to the healthy subjects. The clearance of lepirudin in women is lower by 25% than in men. In elderly patients, the clearance of lepirudin is lower by 20% than in younger subjects, apparently due to the reduction of renal function with age [19].

The anticoagulant effects of lepirudin should be monitored using an activated partial thromboplastin time (aPTT) test, or another appropriate diagnostic test [21], and drug dosage should be adjusted based on the obtained results [19,20,22].

### 3.5. Bivalirudin

Bivalirudin is a specific and reversible direct thrombin inhibitor, which is indicated for use as an anticoagulant in patients with unstable angina undergoing percutaneous transluminal coronary angioplasty (PTCA) [23]. It is a synthetic, 20-amino acid peptide that is administered to the patients as an intravenous (IV) bolus dose followed by IV infusion [24,25].

Similar to lepirudin (see the previous section), following intravenous administration, the time course of bivalirudin plasma concentrations follows a two-compartment model [26]. At the steady state, the drug distribution is confined to the extracellular fluids (V = 14 L in the healthy young subjects).

Bivalirudin is cleared from plasma by a combination of renal mechanisms and proteolytic cleavage, with a half-life of 25 min in patients with normal renal function [27]. The clearance of bivalirudin is dependent on the renal function [24,25,27]. Clearance was similar for patients with normal renal function and mild renal impairment (60–89 mL/min). It was reduced approximately by 20% in patients with moderate to severe renal impairment, by 60% in severe renal impairment, and by approximately 80% in dialysis-dependent patients [27]. Approximately 25% of the drug can be cleared by hemodialysis [23].

The clearance of bivalirudin is not dependent on the administered dose or patients’ gender [27]. Patients’ weights and renal function are the major factors for the selection of bivalirudin dosage, along with the monitoring of the anticoagulation status for dose adjustment [23,27].

### 3.6. Ziconotide

Ziconotide is a synthetic 25-amino acid antagonist of the N-type calcium channel [28]. This drug is delivered intrathecally (IT) and is indicated for the management of severe chronic pain in certain groups of patients [29].

Ziconotide is formulated as a sterile, preservative-free, isotonic solution for IT administration using an appropriate microinfusion device [28]. Following IT administration, the cerebrospinal fluid (CSF) volume of distribution of ziconotide (155 mL) is similar to the estimated total CSF volume [30]. The drug is cleaved in the body by endopeptidases and exopeptidases, and its terminal half-life is approximately 4.6 h [29]. Following IT administration, the ziconotide that leaves the brain and the spinal cord is effectively diluted in the body fluids and exerts low systemic exposure [29].

Due to the special local administration route, the pharmacokinetics of IT ziconotide does not depend on the renal function, weight, gender, or other patient-related factors [28,29,31].

### 3.7. Exenatide (IR and SR Formulations)

Exenatide is a 39-amino acid peptide amide, a glucagon-like peptide-1 (GLP-1) receptor agonist, that acts as an incretin-mimetic agent, enhances glucose-dependent insulin secretion, and exhibits additional antihyperglycemic pharmacological effects [32]. Exenatide is used in the form of twice-daily subcutaneous (SC) injections of immediate release (IR) formulation (drug solution) to the patients with type 2 diabetes.

The absolute bioavailability of IR exenatide following SC administration has not been reported, but apparently it is close to 100%, based on the recent population-based pharmacokinetic modeling analysis of clinical data from eight trials [33]. The drug absorption following SC administration is rapid, and maximal plasma concentrations are attained approximately two hours after the injection. The apparent volume of distribution following SC administration is 28.3 L, and it is dependent on the body weight [33].

Exenatide is predominantly eliminated by glomerular filtration and proteolytic degradation. The mean apparent clearance of exenatide is 9.1 L/h, and the mean terminal half-life is 2.4 h. In patients with mild to moderate renal impairment (creatinine clearance 30 mL/min to 80 mL/min), exenatide clearance is only mildly reduced, and no dose adjustment is necessary [34]. However, a 10-fold reduction in exenatide apparent clearance was observed in patients with end-stage renal disease receiving dialysis (in comparison to the healthy subjects), necessitating a substantial dose reduction. The pharmacokinetics of exenatide is not affected substantially by the hepatic insufficiency [35], gender, race, obesity, and advanced age (in geriatric patients), and these factors do not require a dose adjustment of the drug [36].

Once-weekly (sustained release, SR) exenatide formulation contains the same drug that is encapsulated in the microspheres made of a biodegradable poly (d,l-lactide-co-glycolide) (PLGA) polymer [37]. These microspheres should be resuspended (manually, or using a dual-chamber pen) prior to the SC injection to the patient [38]. Exenatide is released at a constant rate from the injected microspheres; the steady state as attained after six to seven weeks, and the concentrations of the drug in the plasma decrease gradually over three months upon treatment discontinuation [37]. The released drug follows the same distribution and elimination processes as the IR injection [39], and apparently is affected by the same variability factors (body weight for the distribution, and renal function for the clearance; see above).

### 3.8. Lixisenatide

Lixisenatide is a 44-amino acid peptide, a glucagon-like peptide-1 (GLP-1) receptor agonist, with the same pharmacological effects as exenatide [40] (see above). Lixisenatide should be administered once daily as SC injections of immediate release formulation (drug solution).

The absolute bioavailability of lixisenatide following SC administration has not been reported. The drug absorption following SC administration is rapid, and maximal plasma concentrations are attained 1–3.5 h after the injection. The apparent volume of distribution (V/F) following SC administration is approximately 100 L, and it is dependent on the body weight [41].

Lixisenatide is eliminated by glomerular filtration and proteolytic degradation. The mean apparent clearance (CL/F) of the drug is 35 L/h and the mean terminal half-life is three hours. Mild, moderate, and severe renal impairment lead to increased maximal plasma concentrations (C_max_, by 60%, 42% and 83%, accordingly) and area under plasma concentration vs. time curve (AUC, by 34%, 69% and 124%, accordingly) values of lixisenatide [40]. Indeed, creatinine clearance was identified as a covariate for the apparent clearance of lixisenatide based on the population-based pahrmacokinetic modeling analysis [41]. Despite this, no dose adjustment of lixisenatide is recommended in patients with mild and moderate renal impairment [34,40], but close monitoring of drug safety (the GI adverse effects and renal functions) in these patients is advised. Use of the drug in patients with end-stage renal disease is not recommended.

The pharmacokinetics of lixisenatide is not affected substantially by the hepatic insufficiency [35], gender, race, body weight, and advanced age (in geriatric patients) [42], and these factors do not require any dose adjustment of the drug.

### 3.9. Linaclotide

Linaclotide is a 14-amino acid peptide for the treatment of chronic constipation and constipation-predominant irritable bowel syndrome (IBS-C) [43]. Linaclotide acts on the guanylate cyclase-C (GC-C) receptors on the luminal membrane of the GI tract to increase chloride and bicarbonate secretions into the intestine; it also inhibits the absorption of sodium ions, thus increasing the secretion of water into the lumen and improving defecation [44,45,46].

The absorption of linaclotide into the systemic circulation is minimal, and only a few patients that received the regular oral doses of linaclotide had detectable plasma drug concentrations [43]. In the GI tract, linaclotide is metabolized to the active 13-amino acid metabolite, MM-419447 (des-tyrosine) by the removal of the C-terminal tyrosine [47]. After the standard oral doses, concentrations of linaclotide and its active metabolite in plasma are below the limit of quantitation. Therefore, standard pharmacokinetic parameters cannot be calculated for these compounds [43]. Both linaclotide and its metabolite are proteolytically degraded in the intestinal lumen to smaller peptides and natural amino acids.

Neither linaclotide nor its active metabolite are inhibitors or inducers of cytochrome P450 enzymes, or inhibitors of common human efflux or uptake transporters [46,48]. The lack of absorption, local action in the gastrointestinal tract, and proteolytic metabolism mean that age, gender, renal impairment, and hepatic impairment do not alter the pharmacokinetics of oral linaclotide [43,46,48].

### 3.10. Plecanatide

Plecanatide is a 16-amino acid peptide for the treatment of chronic constipation and constipation-predominant irritable bowel syndrome (IBS-C) [49,50,51]. Similar to linaclotide (see the previous section), plecanatide and its active metabolite act locally on the luminal membrane of the GI tract, and are minimally absorbed into the systemic circulation [49,50,51]. Thus, standard pharmacokinetic parameters cannot be calculated for these compounds, and dose adjustment of plecanatide based on the common variability factors is not necessary.

## 4. Concluding Remarks and Perspectives

The clinically approved toxin-derived peptide drugs are very dissimilar in many respects. They originate from different species and affect very different targets at different locations in the body (see Table 1). This necessitates their administration via different routes, including the most common and convenient oral route (for local effects—linaclotide and plecanatide; or for systemic effects—captopril and cyclosporine), and one of the least convenient and invasive routes of administration—an intrathecal infusion mode (for ziconotide).

Based on their chemical structure (see Figure 1), clinically approved toxin-derived peptide drugs can be classified into peptides or peptide-derived drugs (such as captopril); they can also be classified into linear or cyclic compounds, which contain or lack intramolecular disulfide bridges (see Figure 1 and Table 2). Of course, these structural differences have a profound effect on the physicochemical properties of the analyzed drugs, only some of which are summarized in this review (see Table 2).

Molecular weight appears to be one of the most important properties of the analyzed toxin-derived peptide drugs, since it directly affects the suitability of the specific compound for oral administration to the patients, and their subsequent pharmacokinetics. The relatively high molecular weight of linaclotide and plecanatide, combined with their limited water solubility (see Table 2), are beneficial for their local effect within the GI lumen after oral administration. The combination of these properties limits the systemic absorption of linaclotide and plecanatide (see Table 3) that contributes to their selectivity and safety. On the other hand, a similar combination of properties for cyclosporine is undesired, since it limits the absorption of the drug after oral administration and contributes to the high pharmacokinetic variability of this drug (see Table 4).

For the toxin-derived peptide drugs that are administered intravenously and locally (eptifibatide, lepirudin, bivalirudin, and ziconotide), high molecular weight is not a limitation, and can be advantageous, because it limits the distribution of the drug from the site of administration to other locations in the body. Indeed, all of the above-mentioned drugs have small volumes of distribution, and exert their desired pharmacological effect at the administration site. For instance, the IV-administered drugs (eptifibatide, lepirudin, and bivalirudin) have a volume of distribution values in the range of 12–19 L (see Table 3), indicating their distribution in the blood and in the extracellular fluid, and exert their effects within the blood vessels (inhibition of platelet aggregation or of thrombosis; see Table 1). Ziconotide distributes in the brain, which is the site of the desired effect (management of chronic pain; see Table 1), and exerts low systemic exposure following intrathecal infusion.

A small volume of drug distribution, which characterizes many toxin-derived peptide drugs (see Table 3), indicates that these drugs can undergo efficient elimination by the clearing organs (kidney and liver). Indeed, renal elimination is a major pathway of clearance for the analyzed drugs, and renal function-based (CLcr) dosage adjustment is needed for many of them to avoid toxicity (see Table 4). An additional major elimination pathway, which is characteristic for all of the toxin-derived peptide drugs, is proteolysis by proteases that are abundant in different locations in the body. Therefore, proteolysis affects peptide drugs at the sites of their administration and distribution, irrespective of whether their volume of distribution is small or big.

A small volume of drug distribution, combined with efficient clearance via renal and proteolytic routes (see above), results in short half-life values for all of the analyzed toxin-derived peptide drugs (see Table 3). For IV-administered drugs that affect a critical pathway (blood clotting), such a short half-life is advantageous, since it allows a tight control of this pathway by changing the drug infusion rate. On the other hand, a short half-life requires continuous drug infusion under medical supervision, which can be inconvenient to the patient. This drawback is especially pronounced for ziconotide, which is infused intrathecally using special catheters and microinfusion devices.

A short half-life can be disadvantageous also for the subcutaneously-administered anti-diabetic agents, despite prolonged duration of their pharmacological effects that continue also after elimination of the drug from the body. A standard dosing of exenatide and lixisenatide requires twice-daily and single-daily SC injections, which are inconvenient for the patients. To overcome this limitation, once-weekly SR formulation has been developed for exenatide (see Section 3.7 above).

To summarize, the chemical structure of toxin-derived peptide drugs has specific effects on their pharmacokinetics. These drugs tend to have limited permeability via biological barriers, leading to a low volume of their distribution, and the need to administer them invasively (via IV, SC, or intrathecal route). These drugs are eliminated efficiently via the kidneys and proteolytic degradation, leading to their short half-life and the need for frequent or continuous dosing. The dose of many toxin-derived peptide drugs should be adjusted in the case of renal insufficiency, and monitoring patients’ renal function (CLcr) is needed to maintain the desired efficacy/safety balance of these drugs.

There are additional consequences of the chemical structure of toxin-derived peptide drugs that were not covered in this review. For instance, the peptide nature of these drugs is a challenge for their manufacture; it limits the stability (shelf-life) of their drug products, necessitates specific shipment and storage conditions, etc. The pharmacodynamic aspects of toxin-derived peptide drugs were mentioned only in a superficial way in this review. The high potency of these drugs, which stems from their origin (toxins), affects their safety (i.e., the balance of their desired versus adverse effects) the selection of doses, and dose adjustment in the individual patients.

This review focused only on the toxin-derived peptides that were approved for clinical use. Several agents from this group were discontinued in clinical trials stage (e.g., α-conotoxin Vc1.1, χ-conotoxin-MrIA, contulakin-G, conantokin-G, cenderitide and others; these are reviewed by Pennington et al. in [2]). Clinical development of the majority of these peptides was discontinued because of toxicity-related issues, and not because of unfavorable pharmacokinetic properties. Overall, it appears that the chemical structure of a specific toxic peptide can be a useful source for the development of toxin-derived peptide drugs. Exploration of the “chemical space” of a specific toxin and its extensive modification using the available techniques (e.g., by changing the amino acid sequence, cyclization, introducing additional chemical groups, etc.; see Figure 1) can lead to the acceptable (or even desirable) combination of the physicochemical and pharmacokinetic properties for the subsequent pre-clinical and clinical development of toxin-derived drug leads. The toxicity of some toxin-derived agents is expected, and unfortunately cannot be avoided completely at the different stages of drug development.

Overall, toxin-derived peptides are a heterogeneous, interesting, and promising group of agents that can be suitable for many clinical applications. They have specific structural properties that affect their pharmacokinetics and pose some limitations on their clinical use. These properties should be taken into account during the development of new toxin-derived peptide drugs and for the efficient and safe use of the clinically approved drugs from this group in individual patients.

## Figures and Tables

**Figure 1 toxins-10-00483-f001:**
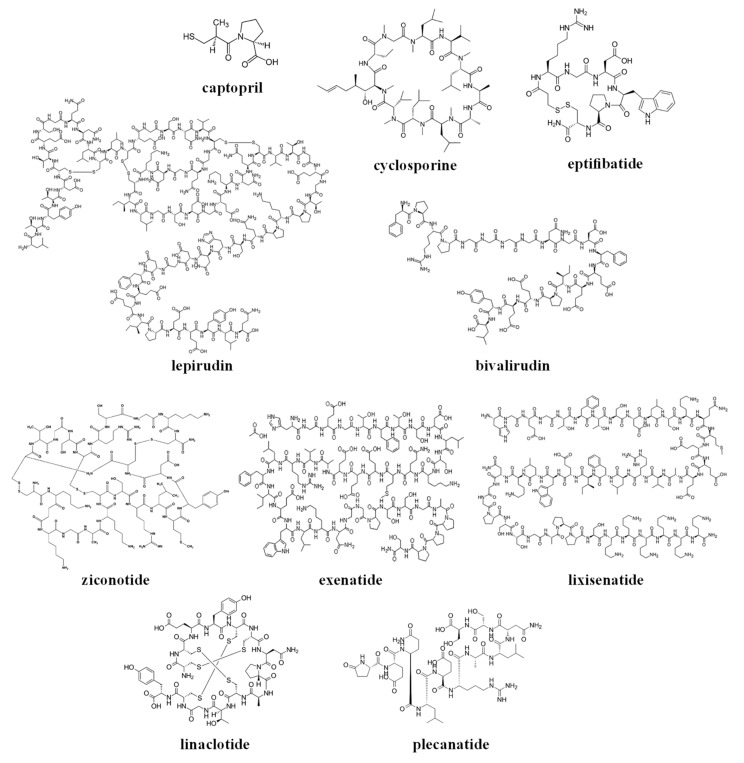
The chemical structures of the clinically approved toxin-derived peptide drugs.

**Table 1 toxins-10-00483-t001:** Summary of the clinically-approved toxin-derived peptide drugs and of their innovative drug products.

Drug	Derives From	Mechanism of Action	Major Indication	Innovator Product/s	Major Doses and Formulation/s	Administration Route	Innovator Company	Approved for Clinical Use in
Captopril	Bradykinin-potentiating factor from the venom of a lancehead viper (Bothrops jararaca)	angiotensin-converting enzyme (ACE) inhibitor	hypertension	Capoten	6.25 mg, 12.5 mg, 25 mg, 50 mg, and 100 mg tablets	PO	Bristol Myers Squibb	1981
Cyclosporine	*Tolypocladium inflatum* fungus	calcineurin inhibitor	immunosuppression	Sandimmune, Neoral	10 mg, 25 mg, and 100 mg solution or capsules	PO	Novartis	1983
Eptifibatide	Pigmy rattlesnake (*Sistrurus miliarius*)	reversible antagonist of the platelet glycoprotein (GP) IIb/IIIa receptor	platelet aggregation inhibition	Integrilin	0.75 mg and 2 mg solution	IV	Merck Ltd.	1999
Lepirudin	European medicinal leech (*Hirudo medicinalis*)	thrombin inhibitor	prevention of thrombosis	Refludan	20 mg and 50 mg solution	IV	Bayer Healthcare Pharmaceuticals	1997 (manufacture terminated in 2012)
Bivalirudin		thrombin inhibitor	prevention of thrombosis	Angiomax	250 mg solution	IV	The Medicines Company	2000
Ziconotide	Magical cone marine snail (*Conus magus*)	selective N-type voltage-gated calcium channel blocker	chronic pain	Prialt	25 ug and 100 ug solution	Intrathecal	Elan Pharmaceuticals	2004
Exenatide	Gila monster lizard (*Heloderma suspectum*)	glucagon-like peptide-1 (GLP-1) receptor agonist	type 2 diabetes	Bydureon, Byetta	5 mg, 10 mg, 250 ug, and 2 mg solution or extended release suspension	SC	Astra Zeneca	2011
Lixisenatide		glucagon-like peptide-1 (GLP-1) receptor agonist	type 2 diabetes	Adlyxin, Lyxumia	10 ug, 20 ug, 50 ug, and 100 ug solution	SC	Sanofi-Aventis	2016
Linaclotide	Heat-stable enterotoxin from the pathogenic *E. coli*	guanylate cyclase-C agonist	constipation	Linzess, Constella	72.5 ug, 145 ug, and 290-ug capsules	PO	Allergan Pharmaceuticals	2012
Plecanatide		guanylate cyclase-C agonist	constipation	Trulance	3 mg immediate release tablets	PO	Synergy Pharmaceuticals Inc.	2017

**Table 2 toxins-10-00483-t002:** The major physicochemical properties of the clinically approved toxin-derived peptide drugs. MW: molecular weight.

Drug	MW	Number of Amino Acids and Structural Features	Physiological Charge	logP	Water Solubility mg/mL
Captopril	217	1, with coordinating sulfhydryl-containing moiety	−1	0.34	4.52
Cyclosporine	1203	11, cyclic	-	3.64	0.04
Eptifibatide	832	7, cyclic	-	-	1
Lepirudin	6963	65, with 3 disulfide bridges	-	-	freely soluble
Bivalirudin	2180	20	−4	−0.76	0.0464
Ziconotide	2639	25, with 3 disulfide bridges	-	-	freely soluble
Exenatide	4187	39	-	-	3
Lixisenatide	4910	44	−6	4.15	6
Linaclotide	1527	14, with 3 disulfide bridges	−1	−1.5	0.701
Plecanatide	1682	16, with 2 disulfide bridges	−8	0.64	0.165

**Table 3 toxins-10-00483-t003:** The major pharmacokinetic parameters of the clinically approved toxin-derived peptide drugs (in a typical adult 70-kg patient with normal renal function).

Drug	Absolute Bioavailability, F	Volume of Distribution, V or Apparent V (V/F) L	f_u_ %	Clearance, CL or Apparent CL (CL/F) L/h	t_1/2_ h	T_max_ h
Captopril	60–75% (PO)	56	65–70	49	2	0.75–1
Cyclosporine	10–89% (variable, PO)	210–350	10	21–29	5–27	1.5–2.0
Eptifibatide	100% (IV)	13–18 (coronary artery disease)15.4–19 (healthy)	75	3.85	2.5	-
Lepirudin	100% (IV)	12.2	-	9.8	1.3	-
Bivalirudin	100% (IV)	14	-	14.3	0.42	-
Ziconotide	50% (intrathecal)	0.155	-	-	2.9–6.5	-
Exenatide	100% (SC)	28.3	-	9.1	2.4	2.1
Lixisenatide	-(SC)	100	45	35	3	1–3.5
Linaclotide	~0% (PO, local activity in the GI, is not absorbed systemically)	-	-	-	-	-
Plecanatide	~0% (PO local activity in the GI, is not absorbed systemically)	-	-	-	-	-

**Table 4 toxins-10-00483-t004:** The major pharmacokinetic variability factors, and their use for dose adjustment of the clinically approved toxin-derived peptide drugs.

Drug	Variability Factor	Dose Adjustment
Captopril	dimerization and interaction with endogenous thiol-containing compounds in the plasma	-
	active tubular secretion in the kidneys	dose reduction in renal insufficiency
Cyclosporine	body weight	therapeutic monitoring of trough blood concentrations dosage adjustment taking into account the variability factors
	co-administration of inhibitors of CYP3A	
	hematocrit	
	and additional factors	
Eptifibatide	renal elimination of the drug	maintain the IV bolus component and reduce the IV infusion component of the dosage regimen in patients with renal insufficiency
Lepirudin	renal elimination of the drug	dose selection based on patients’ weight dose reduction in renal insufficiency monitoring of anticoagulant effects (aPTT test)
	gender, age, and disease state affect the drug distribution and elimination	
Bivalirudin	renal elimination of the drug	dose selection based on patients’ weight dose reduction in renal insufficiency monitoring of anticoagulant effects (aPTT test)
Ziconotide	not reported	-
Exenatide	body weight	-
	renal elimination of the drug	dose reduction in renal insufficiency
Lixisenatide	body weight	-
	renal elimination of the drug	no dose reduction, but close monitoring of drug safety, in mild or moderate renal impairment; use of the drug in patients with end stage renal disease is not recommended
Linaclotide	not reported	-
Plecanatide	not reported	-

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
