# Peer review of "Pharmacokinetics of Toxin-Derived Peptide Drugs"

_toxins, 2018, doi:10.3390/toxins10110483_

Round 1
Reviewer 1 Report
This is an interesting review on a less discussed aspect of toxin based drug discovery. The following concerns should be addressed before this work is acceptable for publication:
Major
This review would benefit greatly from some examples of toxin-derived peptides that failed, due to their specific pharmacokinetic properties, to make it to the drug market. This would contribute to a more comprehensive discussion.
Minor
Table I: indicate the origin of lixisenatide in the table.
Table IV: I do not see the purpose to ad ziconotide, linaclotide and plecanatide to this table if no information is available regarding pharmacokinetic variability factors.
Line 29: “The” should be “they”
Author Response
Dear Reviewer,
thank you for the comments.
Based on your suggestion, I've added a paragraph on the discontinued toxin-derived drug leads to the last section of the revised manuscript (lines 371-382).
The origin of lixisenatide is the same as exenatide (the previous line in the same table). I'll verify that the corresponding cells in the table are merged (during the final proofing of the manuscript).
Based on your comment, I've revised the table and some of the text in Table IV. I believe that this table should list all the drugs (consistent with the previous tables and Figures) and state that information on some drugs is lacking.
Based on your comment, I've corrected this typo error.
Reviewer 2 Report
This review has outlined and analyzed pharmacokinetic properties of toxin-derived peptide drugs. It is a valuable review and gives important insights that can be used for development of new toxin-derived peptide drugs and for efficient and safe use of the clinically-approved drugs. The focus of this review is on the approved drugs for clinical use and not on agents in pre-clinical or clinical trials. It is a first on its type and can guide next reviews when the newer agents make it to the market. Authors have divided the sections of review rationally and covered relationship between chemical structure, physico-chemical properties, and the pharmacokinetics of the drugs. They have also summarized the major pharmacokinetic properties and parameters of these drugs that can describe the major variability factors and extent of their effect on the pharmacokinetics of the individual drugs. It is a well structured and well written review with useful summary tables.
Toxin-derived peptides are a heterogeneous group of agents that can be suitable for many clinical applications. Diverse range of structural properties in this group affect their pharmacokinetics and their clinical use. Knowing the properties can facilitate development of new toxin-derived peptide drugs that can be used efficiently and safely in the clinic.
This article is recommended with no hesitation for further process to publication.
Author Response
Thank you for the review and for the positive evaluation of this manuscript.